

# Recent changes in area and thickness of Torngat Mountain glaciers (northern Labrador, Canada)

N.E. Barrand[1], R.G. Way[2], T. Bell[3], and M.J. Sharp[4]

[1]School of Geography, Earth and Environmental Sciences, University of Birmingham, UK
[2]Department of Geography, University of Ottawa, Canada
[3]Department of Geography, Memorial University of Newfoundland, Canada
[4]Department of Earth & Atmospheric Sciences, University of Alberta, Canada

*Correspondence to:* N.E. Barrand (n.e.barrand@bham.ac.uk)

**Abstract.** The Torngat Mountains National Park, northern Labrador, Canada, contains more than 120 small glaciers; the last remaining glaciers in continental northeast North America. These small cirque glaciers exist in a unique topo-climatic setting, experiencing temperate maritime summer conditions, yet very cold and dry winters, and may provide insights into the deglaciation dynamics of

similar small glaciers in temperate mountain settings. Due to their size and remote location, very little information exists regarding the health of these glaciers. Just a single study has been published on the contemporary glaciology of the Torngat Mountains, focusing on net mass balances during three years in the early 1980s. This paper addresses the extent to which glaciologically-relevant climate variables have changed in northern Labrador in concert with 20th century Arctic warming, and how these

changes have affected Torngat Mountain glaciers. Field surveys and remote-sensing analyses were used to measure regional glacier shrinkage of 27% from 1950-2005, substantial rates of ice surface thinning (up to 6 m yr$^{-1}$) and negative geodetic mass balances at Abraham, Hidden and Minaret Glaciers, between 2005 and 2011. Glacier mass balances appear to be controlled by variations in winter precipitation and, increasingly, by strong summer and autumn atmospheric warming since

the early-1990s. This study provides the first comprehensive contemporary assessment of Labrador glaciers and will inform both regional impact assessments and syntheses of global glacier mass balance.





## 1 Introduction

The glaciers of the Torngat Mountains, northern Labrador, Canada, occupy a unique physiographic
and climatic setting at the southern limit of the Eastern Canadian Arctic. Their proximity to the
Labrador Current provides temperate, maritime summer conditions, yet very cold and dry winters.
Examination of Torngat glacier change may provide insights into the deglaciation dynamics of other
similarly situated small glacier populations (e.g. around the south Greenland ice sheet peripheries),
and other small glaciers retreating into cirque basins in temperate high mountain settings (e.g. Euro-
pean Alps and Pyrenees, North American Rockies and Cascade Mountains). In addition, the glaciers
of the Torngats contribute to the total land ice mass of very small glaciers (e.g. Bahr & Radić, 2012),
are an important component of the local Arctic tundra and fjord ecosystem, and form part of the
cultural landscape of the Labrador Inuit (Brown et al., 2012). Whilst early exploration and recon-
naissance survey flights led to classification and mapping of some Labrador glaciers (e.g. Forbes,
1938; Henoch & Stanley, 1968), a complete inventory and analysis of the glaciers of the Torngat
Mountains, covering a total active glacier area in 2005 of $22.5 \pm 1.8$ km$^2$ (not including relict ice),
has only recently been completed and incorporated into global glacier datasets (Way et al., 2014;
Pfeffer et al., 2014; Arendt et al., 2015). The majority of the glaciological research in the region
has focused on reconstructing Pleistocene palaeoglaciological histories, in particular the timing and
maximum extent of the Laurentide ice sheet during the last (Wisconsinan) glaciation (e.g. Ives, 1978;
Clark, 1988; Clark et al., 2003), and the extent and timing of Little Ice Age (LIA) glacier readvance
(e.g. McCoy, 1983; Rogerson et al., 1986). Combining the new regional glacier inventory (Way
et al., 2014) with field and remote-sensing based analyses, Way et al. (2015) showed a consistent
glacial response to centennial-scale regional climate warming by documenting a 52.5% reduction in
ice extent since glaciers reached their LIA maxima (dated to between 1581 and 1673).

The only other work published on the contemporary glaciology of the Torngat Mountains was
based on field research expeditions conducted by Memorial University, Canada, during the early
1980s (Rogerson, 1986; Rogerson et al., 1986a). Between 1981 and 1984, these authors measured
net mass balances using the glaciological method, from centreline and distributed transects of ab-
lation stakes and snowpits, on clean and debris-covered ice at four cirque glaciers in the Selamiut
Range, south of Nachvak Fjord, in the central part of the National Park (Figure 1). In all three years
between 1981 and 1984, the Abraham and Hidden Glaciers in the adjacent McCornick river valley
experienced negative net balances, ranging from –0.21 to –1.28 m yr$^{-1}$ w.e. Nearby higher-elevation
Minaret Glacier and Superguksoak (at 1.37 km$^2$, the largest glacier in the region) had near-zero
balances in 1981, positive balances in 1982 (both 0.28 m yr$^{-1}$ w.e), and slightly negative balances
(both –0.16 m yr$^{-1}$ w.e) in 1983. These results were linked to annual variations in winter precipi-
tation (Rogerson et al., 1986). Aside from the 2005 inventory (Way et al., 2014), no more recent
information is available regarding the health of these four glaciers, or of the remaining glacial ice in
Labrador. A key question is the extent to which glaciologically-relevant climate variables in north-





ern Labrador have changed in concert with Arctic warming during the 20th century (Kaufman et al.,
2009), and how these changes have affected Labrador glaciers.

This study uses field, remote sensing, and homogenized climate and reanalysis datasets to exam-
ine the response of glaciers in northern Labrador to prevailing climatic conditions since the mid 20th
century. The 2005-dated inventory of Way et al. (2014) (derived from very high-resolution aerial
orthophotos) is used as a basis for historical change assessment since the mid-20th century by com-
paring to glacier outlines derived from archival aerial survey photography (1950). Recent glacier
changes are calculated using glacier areas derived from high-resolution spaceborne electro-optical
imagery (2008). Regional volume and volume change estimates are calculated using volume-area
scaling relations. Field and remote sensing surveys of ice surface topography at three of the four
glaciers visited in the early 1980s are used to calculate rates of surface elevation change between
2005 and 2011, and to calculate geodetic mass balances. Finally, climate and reanalysis datasets are
selected to view both 20th century area changes and 1980s and late 2000s glacier mass balances
in the context of prevailing climatic trends. This study provides the first assessment of the contem-
porary state of Labrador glaciers and their likely climate sensitivities, the results of which may be
incorporated into both regional impact assessments and syntheses of global glacier mass balance.

## 2  Study Area and Methods

The Torngat Mountains are the southernmost mountain range in the eastern Canadian Arctic, occu-
pying a position towards the north of the Labrador Peninsula, south of Baffin Island (Figure 1). The
synoptic climatology of the region is strongly influenced by the Labrador Current which transports
cold Arctic water into the Labrador Sea and southward along the Labrador coast (Myers & Donnelly,
2008). The influence of the Labrador Current produces maritime summer conditions and colder, drier
winters due to extensive seasonal sea-ice formation (e.g. Kvamstø et al., 2004). Precipitation during
winter is generated by storms that are directed by the influence of the Canadian Polar Trough on
the position of Arctic and Polar fronts (Moore et al., 2011). Glaciers in the Torngat Mountains are
preserved due to their proximity to this coastal moisture source, as well as their topographic setting
(reducing incoming solar radiation receipt) and, in some cases, the insulating effect of supraglacial
debris (Rogerson, 1986; Rogerson et al., 1986a; Way et al., 2014). The four glaciers measured in the
early 1980s are influenced by a combination of these factors. Abraham and Hidden Glaciers (Figure
2; labelled A,H) are less than 20 km from the coast, have partly debris-covered termini, and are lo-
cated below ∼500 m-high backwalls comprising Cirque Mountain (1568 m above sea level (a.s.l.)).
Superguksoak Glacier (Figure 2, labelled S) is similarly situated, and is debris-covered over ∼40%
of its surface area. Minaret Glacier (Figure 2, labelled M) does not have extensive supraglacial debris
cover, but is located ∼350 m higher than the other glaciers, and is shaded by a large backwall com-
prising Mount Caubvick (1652 m a.s.l.). The interplay of altitude, supraglacial debris, slope, aspect,




and height of (and proximity to) cirque backwalls has been shown to influence the spatial variability
of surface mass balance (Rogerson, 1986).

### 2.1   Derivation of Area Changes

A glacier inventory derived from late-summer (August) 2005 Parks Canada 1:40,000-scale colour
aerial photographs was used as reference for documenting historical and late 2000s glacier area
changes. Full details of the 2005 inventory data collection, image processing, ice area delineation
and error assessment are provided by Way et al. (2014). Historical glacier outlines were manually
digitized from 107 scanned diapositives of Royal Canadian Air Force survey photographs, acquired
during a two-week period in August, 1950 (LAB Series 47, 48, 50, 79, 91–95). 1:50,000-scale dia-
positives scanned at 1200 dpi provided digital files with a corresponding ground pixel resolution of
~1 m. Image orthorectification in the absence of accurate camera calibration information was un-
dertaken using a rational functions model in PCI Geomatica software (v.10.3). The model calculates
correlations between image pixels and ground locations using a ratio of polynomial functions with
coefficients calculated based on the number and quality of ground control points (GCPs). Between
15 and 20 GCPs per image frame were collected from 2005 orthophotos and an undated 18-m res-
olution Parks Canada digital elevation model (DEM), to correct for topographic distortion due to
terrain relief (e.g. Kääb, 2005). The resultant root mean square (RMS) error of 1950 orthophotos
was <5 m across all images.

To examine mid to late 2000s area changes, scenes from the SPOT5-HRG (High Resolution Geo-
metric) image archive were analysed (Table 1). The very small size of Labrador glaciers and the need
to obtain both cloud and snow-free scenes during summer precluded the use of medium-resolution
spaceborne visible and near-infrared sensor archives such as ASTER and Landsat. SPOT5-HRG im-
ages at 10 m ground resolution in multispectral mode (3 bands between 0.50 and 0.89 $\mu$m), and
5 m in monospectral (panchromatic) mode (0.51-0.73 $\mu$m) provide sufficient spatial resolution to
accurately delineate Labrador glacier outlines following pan-sharpening (e.g. Paul & Kääb, 2005;
Racoviteanu et al., 2008). Cloud- and snow-free scenes used in this work were acquired during late
July and early August 2008 (Table 1).

Glacier extents were delineated by overlaying 2005 outlines onto 1950 and 2008 imagery sets and
editing where detectable changes had occurred (e.g. Figure 2). Due to missing or poor-quality 1950
photo frames, outlines of 101 glaciers were digitized (out of a total of 124 active glaciers in the 2005
inventory). The areas of these 101 glaciers represent 81% of the 2005 total ice area. Due to the large
swath coverage of SPOT5-HRG scenes (60 x 60 km, continuous tiles), all 124 active glaciers in
the 2005 inventory were successfully identified in the 2008 imagery and redigitized. Glacier outline
adjustment (mapping) took place following the protocols of the Global Land Ice Measurements
from Space (GLIMS) initiative (Racoviteanu et al., 2009). Indistinct and debris-covered ice margins
were mapped using morphological features such as lateral meltwater streams, debris flow bands and





changes in surface slope (facilitated by 3-dimensional visualisation of orthophotos draped over the Parks Canada regional DEM) (e.g. Paul & Kääb, 2005; Racoviteanu et al., 2009; Way et al., 2014).

Following the approach of Paul et al. (2013), the error (precision) of glacier area delineations was determined by independent remapping of ice margins. Three separate operators mapped the extent of 10 of the largest glaciers in the region (∼15% of the total ice area), selected to include a representative range of margin characteristics (bare ice, snow-covered, debris-covered, shadowed and unshadowed). The RMS difference between operator extents (0.009 and 0.019 km$^2$ for 1950 and 2008 imagery) corresponds to mapping errors of ∼2 m and ∼5 m (equivalent buffer widths) along respective glacier margins. Buffers of ±5 and ±10 m were subsequently assigned at 1950 and 2008 ice margins to account for cumulative errors in mapping, image planimetry and orthorectification (e.g. Bolch et al., 2010; Way et al., 2014).

### 2.2 Regional Ice Volume Estimation

Volumes of individual glaciers in 1950, 2005 and 2008 were calculated and summed regionally, using the volume-area ($V$-$A$) power law relation: $V = c_a A^\gamma$ (e.g. Chen & Ohmura, 1990; Bahr et al., 1997). Aggregate volumes were calculated for ensembles of glaciers, as errors can be large when reported at the individual glacier scale (e.g. Chen & Ohmura, 1990). Bahr et al. (2015) note that a 1-sigma error around a mean of the multiplicative scaling parameter $c_a$ would produce a 34% error in calculated ice volume. However, volume errors calculated by $V$-$A$ scaling for regional-scale aggregate ensembles of glaciers are typically <25% (Meier et al., 2007). In the absence of independent measurements of glacier volume from which to derive local scaling coefficients, $c_a$ was assigned equal to 0.191 m$^{(3-2\gamma)}$, and the scaling exponent was fixed to the theoretical constant $\gamma$ = 1.375. These values are based on worldwide means of scaling parameters, and a theoretical derivation of the scaling exponent (Bahr et al., 1997, 2015).

To account for variations in the scaling parameters due to region, glacier, or other factors (spatial or temporal), uncertainties in volume (errors) were estimated by calculating volume sensitivities to the choice of scaling coefficients (e.g. DeBeer & Sharp, 2007; Barrand & Sharp, 2010). These coefficients were derived empirically from direct volume measurements of glaciers in the Alps, Cascades, and similar regions (Chen & Ohmura, 1990), of small glaciers in the Canadian Cordillera (DeBeer & Sharp, 2007), and from purely physical considerations (Bahr et al., 1997). Total volumes were calculated with the scaling parameters (units m$^{(3-2\gamma)}$) / exponents (dimensionless): 0.206 / 1.360, 0.115 / 1.405, and 0.210 / 1.360. The maximum difference between calculated volumes was then taken as the measure of uncertainty (error) in total volume calculations (e.g. DeBeer & Sharp, 2007; Barrand & Sharp, 2010). This approach is considered preferable to a component error propagation calculation, as uncertainty in volume calculations due to the choice of scaling coefficients is likely to be an order of magnitude larger than those from the principal individual error sources (e.g. area measurement errors).




### 2.3 Ice Surface Elevation Changes

Field surveys of glacier centreline ice surface topography were undertaken at Abraham and Hidden
Glaciers during August 2008, 2009 and 2011, and at Minaret Glacier during August 2008 and 2009,
using dual frequency (L1 and L2) Global Positioning System (GPS) instruments. Base-station data
were collected at 1 Hz using tripod-mounted Leica GPS 500 receivers fixed over two survey mark-
ers drilled into exposed bedrock in the forefields of Hidden / Abraham and Minaret Glaciers (for
locations see Figure 2). Base-station survey positions were calculated using precise point position
(PPP) processing of 24 hours of positional data, using the Canadian Spatial Reference System On-
line Global GPS Processing Service (CSRS-PPP). Survey positions were corrected from the total
height of the antenna phase centre (including vertical offset). Ice surface topography was measured
at accessible points along glacier centrelines at ∼5 m horizontal spacing using an additional GPS
500 receiver in rover mode connected to a survey-pole mounted antenna. Simultaneous base-station
positions were recorded during surveying to provide differential processing capability. Rover sur-
vey points were differentially corrected (including for pole height) using post-processing tools in
Leica GeoOffice software. Following differential correction, positions with three-dimensional accu-
racy <10 cm were excluded from further analysis. This procedure resulted in elimination of ∼15%
of survey points at higher elevations proximal to cirque backwalls due to the poorer configuration of
available satellites during surveying in those areas.

Elevation changes ($\delta h$) were calculated by differencing individual crossover spot heights ($h$)
within a 1 m radius of the preceding year's survey position. Corrections for along- and across-track
slope were deemed unnecessary due to the small crossover window size and shallow surface slopes
of the accessible (and thus surveyed) parts of the ice surface (typically <5°). Whilst some surface
elevations were excluded from the difference analysis due to positional deviations from the cen-
treline track, sufficient crossovers remained to adequately characterise centreline elevation change
without the need for interpolation or plane-fitting (e.g. Moholdt et al., 2010). Errors in individual
crossover elevation changes were determined empirically by calculating the root sum of squares
(RSS) of errors in receiver position. In addition, three-dimensional GCPs collected in 2008 over
bare rock surfaces around each glacier were used to re-process 10 (Abraham and Hidden) and 5 m-
resolution (Minaret) DEMs, from 2005 Parks Canada air photos. Ice surface elevation changes were
then calculated between 2005 DEM cell pixels and 2008 GPS survey positions.

### 2.4 Geodetic Mass Balance

Centreline ice surface elevation changes were extrapolated using glacier hyposometres to calcu-
late geodetic mass balances from 2005-2008, 2008-2009 (all), and from 2009-2009 (Abraham and
Hidden glaciers only). Hypsometries were extracted from 10 m (Hidden and Abraham) and 5 m
(Minaret) -posting digital elevation models of each glacier surface. DEMs were derived from stereo-





photogrammetric processing of 2005 aerial survey images (Way et al., 2014), Individual centreline elevation changes ($\delta h$) were averaged into 10 m elevation bins and applied to all pixels in each bin across the glacier surface (e.g. Barrand et al., 2010; Nuth et al., 2010). In the event that no $\delta h$

measurements were available (due to safe access issues during surveying or culling of low accuracy points, see section 2.3), that bin was assigned the averaged value of the next adjacent bin. Total volume changes were then calculated by multiplying $\delta h$ by the pixel area and summing across the glacier surface (e.g. Barrand et al., 2009, 2010; Nuth et al., 2010). Volume changes were converted to geodetic mass balances in water equivalent units (m yr$^{-1}$ w.e.) by dividing by the glacier area, mul-

tiplying by the ratio of the density of ice to water ($p_w/p_i = 0.918$, as glaciers were snow-free during surveying (Bader, 1954; Cuffey & Paterson, 2010)) and, in the case of 2005-2008 and 2009–2011 measurements, dividing by the time between epochs.

Uncertainties in volume change and mass balance were calculated following an approach similar to Nuth et al. (2010). Individual point elevation change errors ($E_{PT(\delta h)}$) were calculated empiri-

cally from errors in DEM vertical accuracy or GPS receiver position (section 2.3). Total elevation change errors, $E_Z$, were then calculated by combining $E_{PT(\delta h)}$ in quadrature with extrapolation errors ($E_{EXT}$), reducing by the square root of the total number of independent measurements (e.g. Arendt et al., 2006; Barrand et al., 2010; Nuth et al., 2010), considered to be one measurement per 10 m elevation bin, considering the presence of weakly positive spatial autocorrelation. $E_{EXT}$ quan-

tifies the uncertainty in applying centreline $\delta h$ measurements to entire elevation bins. Following Nuth et al. (2010), $E_{EXT}$ was approximated by the standard deviation of the averaged $\delta h$ in each elevation bin. Where bins had fewer than five $\delta h$ measurements, $E_{EXT}$ was set to twice the glacier average, while bins with extrapolated measurements were assigned three times the average. Total volume change errors, $E_{VOL}$, were calculated as the RSS of $E_Z$ multiplied by the 2005 glacier area.

As all elevation measurement surveys were conducted in the same month of each year, no additional uncertainties were assigned as a result of seasonal changes.

## 3 Results and Discussion

### 3.1 Regional Area and Volume Changes

During the three years from 2005 to 2008, the total area of all 124 glaciers in Labrador shrank from

22.46 to 21.80 km$^2$, a reduction of 0.66 $\pm$ 0.41 km$^2$, or 3% of the 2005 ice area. Due to missing photo frames from the 1950s LAB series, only 101 of the 124 glaciers examined in 2005 could be digitized to provide historical glacier areas. In 1950, these 101 glaciers occupied an area of 27.17 km$^2$, shrinking to 19.78 km$^2$ in 2005, then 19.26 km$^2$ in 2008; reductions of 7.39 $\pm$ 0.65 km$^2$, or 27% from 1950-2005 (–0.49% yr$^{-1}$), and 0.52 $\pm$ 0.33 km$^2$, or 1.37% between 2005 and 2008 (–

0.46% yr$^{-1}$). The 27% ice area loss from 1950-2005 is larger than that from similarly sized glaciers between 1975 and 2000 on nearby southern Baffin Island (–17.4% for glaciers <1 km$^2$) (Paul &





Svoboda, 2009). While the total Labrador glacier area loss rates are similar between periods, examination of individual annually-averaged glacier losses reveals a wide range of rates of change, both within and between measurement periods (Figure 3). Between 1950 and 2005, Labrador glaciers
lost between 0.2 and 1.4% yr$^{-1}$ of their 1950 areas, with glaciers of all sizes decreasing in extent (no glacier increased in size during this period, Figure 3a). From 2005-2008, 26 of the 101 glaciers (26%) experienced zero or negligible area losses within calculated uncertainties, yet ∼30% of the sample experienced strongly accelerated area losses (in excess of 1% yr$^{-1}$, and in several instances, exceeding 5% yr$^{-1}$, Figure 3b). These accelerated area losses between 2005 and 2008 are, however,
subject to uncertainties of ± 0.6 to ± 3% yr$^{-1}$.

Scaling from 101 individual glacier areas, the total volume of ice in Labrador in 1950 was 0.64 ± 0.07 km$^3$. The scaled ice volumes from 124 glaciers of the complete inventories of 2005 and 2008 were respectively, 0.48 ± 0.06 km$^3$, and 0.47 ± 0.05 km$^3$. Comparison of the 101 glaciers present in all three inventories showed volume changes of –0.21 ± 0.07 km$^3$ between 1950 and 2005, and
–0.01 ± 0.04 km$^3$ between 2005 and 2008. Between these two most recent years, uncertainties from the maximum difference between scaling coefficients (section 2.2) were larger than scaled volumes as a result of the relatively small changes in glacier area. Using the average of the old and new glacier areas, the total volume change between 1950 and 2005 corresponds to an area-averaged long-term thinning rate of –0.16 ± 0.05 m yr$^{-1}$ (from volume-area scaling).

## 3.2   Selamiut Range / Cirque Mountain Glaciers

Labrador's largest and only previously-studied glaciers are situated in the Selamiut Range and proximal to Cirque Mountain, south of Nachvak Fjord (Figures 1, 2). Focusing on the four sites visited in the early 1980s, Superguksoak Glacier (the largest glacier in Labrador) decreased in size from 1.47 ± 0.08 km$^2$ in 1950, to 1.37 ± 0.03 km$^2$ in 2005, then further to 1.35 ± 0.11 km$^2$ in 2008. Hidden
and Abraham Glaciers shrank from 0.74 ± 0.04 km$^2$ and 0.61 ± 0.03 km$^2$ in 1950, to 0.65 ± 0.03 and 0.54 ± 0.02 km$^2$ in 2005, then to 0.64 ± 0.04 and 0.53 ± 0.04 km$^2$ in 2008. Finally, Minaret Glacier reduced in size from 0.92 ± 0.05 km$^2$ in 1950, to 0.73 ± 0.02 km$^2$ in 2005, and then 0.71 ± 0.05 km$^2$ in 2008 (Figure 2). Although no area increases were observed, it is important to note that individual year measurements may mask considerable inter-annual variability in glacier area change
(including glacier growth resulting from positive mass balance years) (e.g. Rogerson, 1986).

Hypsometries derived from 2005 stereo-photogrammetric DEMs reveal the area-altitude distributions of Abraham, Hidden and Minaret Glaciers (Figure 4). Abraham and Hidden have bimodal hypsometric distributions, with peaks at ∼800 and 900 m a.s.l. at Abraham, and ∼875 and 975 m a.s.l at Hidden Glacier. All of the ice of Minaret Glacier exists at higher elevations than Hidden and
Abraham, predominantly between 1250 and 1350 m a.s.l. Rates of ice surface elevation change varied between glaciers, and particularly between measurement intervals (Figure 4). From 2005-2008, Abraham, Hidden and Minaret glaciers thinned at average rates of –0.98, –2.32, and –0.93 m yr$^{-1}$.





Between 2008 and 2009, the largest thinning rates were measured at Abraham Glacier (up to 6 m yr$^{-1}$ at 850 m a.s.l.). These very large thinning rates decreased further up-glacier, yet remained around 1 m yr$^{-1}$ at higher elevations. Hidden Glacier, which has a similar area-altitude distribution to Abraham yet is more extensively shaded and debris-covered, experienced slightly lower thinning rates, up to 4 m yr$^{-1}$ at lower elevations, and 1–2 m yr$^{-1}$ above ~860 m a.s.l. This elevation-dependant control on valley glacier elevation change was also observed between 2005 and 2008 at all glaciers, and between 2008 and 2009 at Minaret Glacier, with the largest changes (~4 m yr$^{-1}$) again at lower elevations (1200 m a.s.l.), decreasing to 1–2 m yr$^{-1}$ above ~1250 m a.s.l. Thinning rates at Abraham and Hidden Glaciers were greatly reduced during the period 2009–2011. While relatively few measurements of elevation change were available at Abraham Glacier, they provided an average thinning rate of 0.47 ± 0.22 m yr$^{-1}$. An elevation-dependent thinning rate was observed at Hidden Glacier between 2009 and 2011, decreasing from 1 m yr$^{-1}$ at lower elevations, to 0.2 m yr$^{-1}$ at 1000 m a.s.l. (Figure 4). The large thinning rates measured at Abraham, Hidden, and Minaret glaciers between 2008 and 2011 far exceed the area-averaged long-term (1950-2005) thinning rate from volume-area scaling (–0.16 ± 0.05 m yr$^{-1}$).

The thinning rates observed between the summers of 2005, 2008, 2009 and 2011 correspond to negative geodetic mass balances at Abraham, Hidden and Minaret glaciers between 2005 and 2011 (Figure 7). Strongly negative balances between 2005 and 2008, and 2008 and 2009 coincided with nonexistent or very small accumulation areas at these glaciers, as evidenced by the snow-free summer ice conditions present in 2005 aerial photographs, and during 2008 and 2009 field seasons. At Abraham Glacier, geodetic mass balances were –0.73 ± 0.41, –1.76 ± 0.39, and –0.37 ± 0.35 m yr$^{-1}$ w.e., between 2005-2008, 2008-2009, and 2009-2011. For the same time periods, balances at Hidden Glacier were –1.90 ± 0.15, –1.21 ± 0.21, and –0.24 ± 0.17 m yr$^{-1}$ w.e., and at Minaret Glacier –0.27 ± 0.11 and –1.51 ± 0.25 m yr$^{-1}$ w.e. (2005-2008 and 2008-2009 only) (see Figure 7). The larger uncertainties associated with Abraham and Minaret Glacier balances reflect the relative paucity of available elevation change measurements at these sites, largely a result of safe access issues during surveying at lower elevations and poor GPS satellite coverage within the upper cirque basins.

While geodetic balances are available for years between 2005 and 2011 only, they may be compared with net balances from field measurements obtained in the early 1980s. Net balances at Abraham Glacier in the balance years 1981, 1982 and 1983 were –0.27, –0.46 and –1.28 m yr$^{-1}$ w.e. (Rogerson, 1986). The geodetic balance measured between 2008 and 2009 (–1.76 ± 0.39 m yr$^{-1}$ w.e.) is thus the most negative mass balance yet recorded at this glacier. The 2005-2008 geodetic balance at neighbouring Hidden Glacier (–1.90 ± 0.15 m yr$^{-1}$ w.e.) is more negative than any balance year during the early 1980s (1981, 1982, 1983: –0.24, –0.21 and –0.81 m yr$^{-1}$ w.e.), and is the most negative balance yet recorded at this or any other glacier in Labrador. While these six years of discontinous measurements represent the only geodetic mass balance observations yet recorded at





Labrador glaciers, it is important to note that they may mask considerable interannual variability in
accumulation and ablation at the sampled glaciers and within the wider regional population.

### 3.3 Climatological Drivers

Two gridded observation-based climate datasets and two reanalysis products were selected to provide
a northern Labrador climatological context for the late 20th and 21st century glaciological observa-
tions. Seasonal 2 m air temperatures and precipitation totals (where available) were extracted from
the homogenized observational datasets CanGRID and Berkeley Earth (Zhang et al., 2010; Rohde
et al., 2013), and from the NCEP/NCAR R1 and JRA-55 reanalysis products (Kalnay et al., 1996;
Kobayashi et al., 2015). To remove outlying In order to assess precision (repeatibility) between data
products, a comparative analysis. An intercomparison of these observational data and reanalyses is
worthwhile as both approaches and products have data coverage over northern Labrador yet utilize
different input data and interpolation schemes. They may also vary in time period, domain and spa-
tial resolution. A more detailed overview of widely-used surface-based and reanalysis datasets for
temperature and precipitation trend analysis in the wider Canadian Arctic is provided by Rapaić
et al. (2015). Following Rapaić et al. (2015), Berkeley Earth, NCEP R1 and JRA-55 temperature
anomalies were evaluated for the Torngat Mountains region (58°N-60°N, 295°E-297°E) using the
CanGRID dataset as reference. The strongest regression model fit was found between CanGRID and
Berkeley Earth temperature anomalies ($r^2$=0.96, Figure 5a). This was perhaps to be expected as the
Berkeley Earth product is also observation-based. The best model fit between CanGRID and reanal-
ysis temperature anomalies was JRA-55 ($r^2$=0.89), with noticeably less spread and a better model
fit than NCEP R1 ($r^2$=0.73, Figure 5b,c). A time-series plot of summer (JJA) 2-m air temperature
anomalies in the Torngat Mountains from all four products confirmed the 20th century Arctic sum-
mer warming trend (mean of 0.85°C between 1990 and 2011) (e.g. Kaufman et al., 2009; Rapaić et
al., 2015), yet showed a clear cold temperature bias in NCEP R1 since the mid-1990s (Figure 5d).
Due to this bias and their stronger regression model fits to CanGRID reference data, Berkeley Earth
(global availability) and JRA-55 (including precipitation) were selected for further examination of
seasonal climate indices.

Time series plots of seasonal 2 m air temperatures (JJA, SON, DJF, MAM) and the preceding-
years total accumulation season precipitation (SON+DJF+MAM) were created to examine seasonal
climatic conditions in the Torngat Mountains for the period associated with glacier change measure-
ments (1950-2015). Seasonal temperatures were most similar between Berkeley Earth and JRA-55
datasets in Autumn (Figure 6b), and showed similar inter-annual variability, yet 1-2°C magnitude
temperature difference for summer, winter and spring seasons (Figure 6a,c-d). Each of the time-
series showed long-term warming trends, superimposed with inter-annual and pentad-scale variabil-
ity, consistent with observations throughout the rest of the province of Labrador (Finnis & Bell,
2015; Way & Viau, 2015). Warming trends were most pronounced during summer and autumn af-





ter 1990 (Figure 6a-b). Winter temperatures decreased between 1950 and 1990, before rising again post-1990 (Figure 6c). The post-1990 warming may in part explain individual years of negative mass balance in the late 2000s. Total ablation season precipitation from JRA-55 reanalysis shows several years of above average precipitation between 1974 and 1986, then a slightly increasing precipitation

trend since the early 1990s (Figure 6e). Year-to-year variability in climate-mass balance linkages are presented in Figure 7.

While these climate data are both temporally and spatially averaged and thus may not be representative of climate conditions at individual glaciers, it is possible to examine seasonal temperature and precipitation trends in order to suggest potential drivers of glacier change. Glaciers in Labrador lost

27% of their surface area between 1950 and 2005. As both Berkeley Earth and JRA-55 show largely stable seasonal temperatures between 1950 and the early-1990s, it may be suggested that a majority of this area loss occurred after 1990. Pentadal means of seasonal air temperatures from both records show summer and autumn warming of around 2°C by 2010-2015, compared to pre-1995 levels (Figure 6a,b). This summer warming suggests greater energy available for ablation season melt, while

warmer autumn temperatures (which include several seasonal average temperatures above 0°C) may also indicate an increasing proportion of winter precipitation falling as rain.

The field mass balance measurements of Rogerson (1986), which included years of positive mass balance, were explained by the author in terms of variability in winter snowfall; a finding supported by the precipitation data presented here. JRA-55 precipitation totals during 1976-1985 pentads were

∼8% and 12% greater than the preceding pentadal period (1971-1975) (Figure 6e). The only year of negative mass balance at all four glaciers (1983, mean net balance of –0.60 m yr$^{-1}$ w.e.) coincided with a very low precipitation total during the preceding accumulation season (462 mm). Figure 7 shows 2005-2011 geodetic mass balance results alongside JRA-55 seasonal temperature and precipitation trends. These plots show less negative balances at Abraham and Minaret glaciers between

2005-2008, following a high precipitation total of 726 mm in 2006. Geodetic balances more negative than –1 m yr$^{-1}$ w.e. at all glaciers between 2008 and 2009 followed a low precipitation total of 452 mm during the 2007/2008 winter (which may explain the 2008 snow-free summer ice conditions), and summer temperatures more than 1°C warmer than the 2005-2010 pentadal mean (itself, 1°C higher than the 1950-2015 mean) (Figure 6). The lower thinning rates and less negative geode-

tic balances at Abraham and Hidden glaciers between 2009 and 2011 follow precipitation totals some 17% and 22% greater than the preceding year (2008) accumulation season total (Figure 6e), and an extended ablation season with melting conditions extending into the September-November period (Figure 7). While only tentative climate-mass balance linkages may be drawn from these limited data, inter-annual variability in winter precipitation and, increasingly, post-1995 climate

warming play important roles in contemporary Labrador glacier mass changes (similar to many other northern-hemisphere mountain glacier systems). Longer mass balance records with improved





temporal resolution and more local-scale climatological observations are required to confirm these
findings.

## 4   Summary and Conclusions

This study presents the first comprehensive analysis of the contemporary state of small glaciers in the
remote Torngat Mountains of northern Labrador, Canada. These glaciers occupy a unique position
as the most southerly ice masses in the eastern Canadian Arctic. Very little was previously known
about the recent status of Labrador glaciers, with a single study published on the mass balance of
four glaciers during the early 1980s. Due to their very small sizes and remote locations, analysis of

Labrador glacier change required an experimental design comprising both detailed field observations
and high-resolution remote-sensing image analysis. Glaciers in Labrador lost 27% of their surface
area between 1950 and 2005, a larger proportional loss than that from similarly-sized glaciers in
nearby southern Baffin Island over the same period. They then shrunk a further 3% to $21.80 \pm 2.2$
$km^2$ by 2008, with an associated (scaled) total ice volume of $0.47 \pm 0.05$ $km^3$. Three cirque glaciers

measured in the early 1980s and revisited from 2008-2011 thinned at rates between 0.5 and 6 m $yr^{-1}$
at lower elevations at measurement periods between 2005 and 2011, mostly in excess of the long-
term (1950-2005) thinning rate derived from volume-area scaling ($-0.16 \pm 0.05$ m $yr^{-1}$). These very
high thinning rates resulted in consistently negative geodetic balances at all three glaciers between
2005 and 2011, with a balance of $-1.90 \pm 0.15$ m $yr^{-1}$ w.e. at Hidden Glacier between 2005-2008

being the most negative mass balance yet recorded at this or any other glacier in Labrador. Both
geodetic balances between 2005 and 2011 and net balances from early 1980s field measurements
were linked to prevailing climatic conditions from gridded observational and reanalysis-based cli-
mate datasets. These findings suggested that Labrador glacier mass trends are controlled by vari-
ability in winter precipitation, but are increasingly being influenced by strong summer and autumn

atmospheric warming since the early-1990s.

*Acknowledgements.* Access to SPOT5-HRG scenes, digital orthophotos, diapositives and prints was provided
by Parks Canada, the National Air Photo Library (NAPL, Ottawa), and the National Hydrological Research
Institute (NHRI, Saskatoon). We thank the staff and employees of Parks Canada for access and logistical sup-
port, and J. King, P. LeBlanc, A. Tuglavina and E. Merkuratsuk for assistance in the field. ASTER GDEM

is a product of METI and NASA and is available for download from http://gdem.ersdac.jspacesystems.or.jp.
Climate data were accessed via the University of Maine, USA, Climate Change Institute (CCI) Climate Re-
analyzer: http://cci-reanalyzer.org. This work was supported by grants to TB and MS from the Government of
Canada International Polar Year Program. We thank the editor Etienne Berthier, for comments which improved
the manuscript.





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





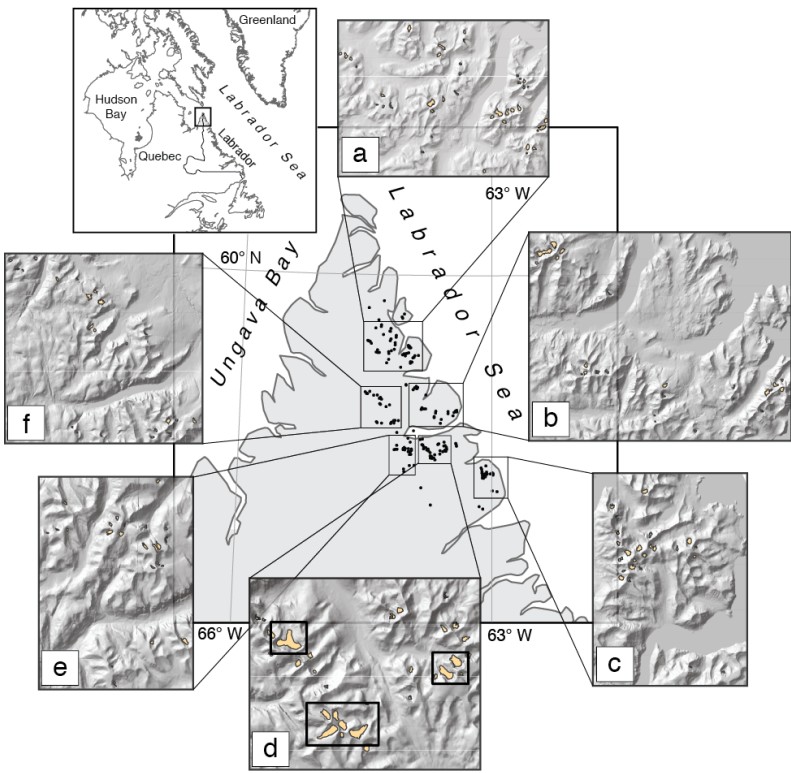

**Figure 1.** Locations (dots in main map) and outlines of Torngat Mountain glaciers, Labrador, Canada (upper-left inset). Local glacier populations include: a) Four Peaks / Ryan's Bay; b) Mount Tetragona / Mount Eliot / Razorback Mountain; c) Blow Me Downs; d) Selamiut Range / Cirque Mountain; e) Tallek Arm; and f) Komaktorvik Lakes. Inset map shaded relief elevations are derived from ASTER Global DEM data. Outlines in panel d refer to glacier locations in Figure 2.





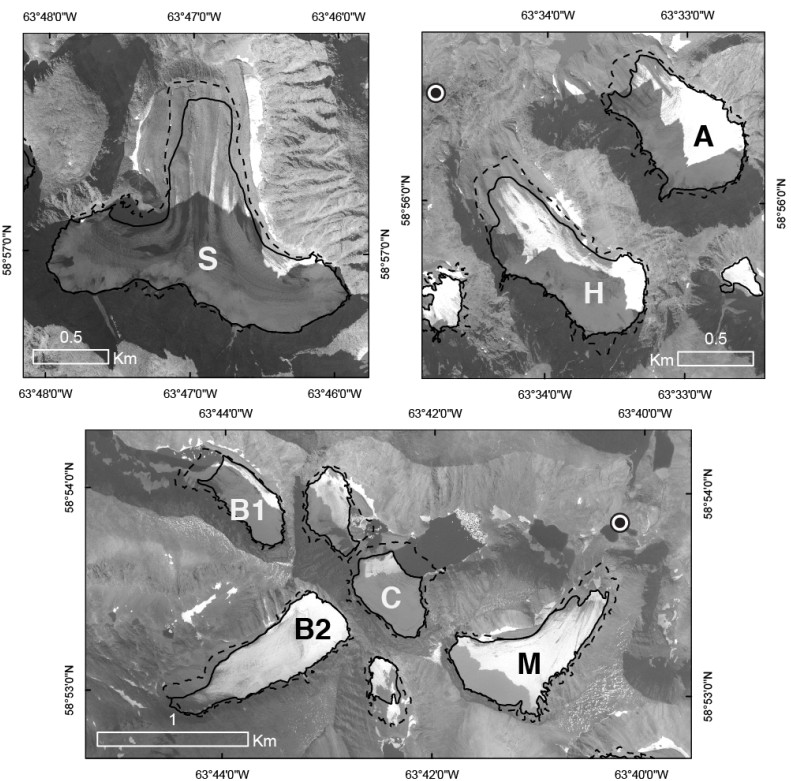

**Figure 2.** Areal extent of glaciers in the Selamiut Range and Cirque Mountain (see Figure 1), in 1950 (dashed lines) and 2005 (solid lines). Background images are 1 m resolution Parks Canada orthophotos acquired in August 2005 (frame numbers 5026_43_4, upper left; 5022_43_136, upper right; and 5022_41_166, lower panel). Text labels refer to glacier names, as follows: S - Superguksoak; H - Hidden; A - Abraham; B1 - Bikini 1; B2 - Bikini 2; C - Caubvick; M - Minaret. Bullseye symbols denote locations of GPS basestations for topographic surveys.



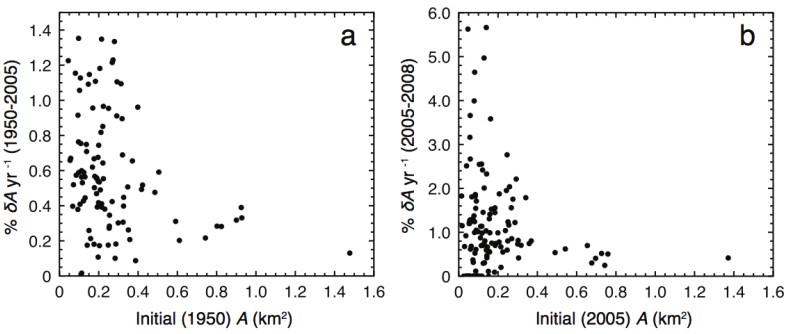

**Figure 3.** Annually averaged percentage glacier area loss ($\delta A$) plotted against initial glacier area (in km$^2$), between 1950 and 2005 (a), and from 2005 to 2008 (b). Note difference in *y*-axis scale.

**Table 1.** SPOT5-HRG scene identification numbers and dates of image acquisition.

| Scene ID | Date |
|---|---|
| 56292290807161536431A0 | 16/07/2008 |
| 56332300807211540311A0 | 21/07/2008 |
| 56302290807251603262A3 | 25/07/2008 |
| 56302280808101555161A3 | 10/08/2008 |
| 56292280808111536031A0 | 11/08/2008 |



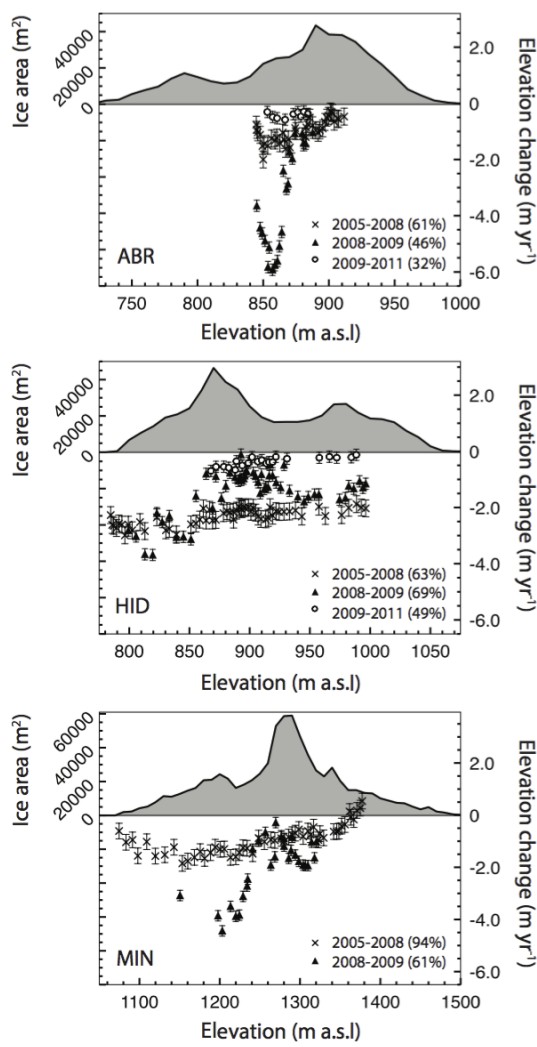

**Figure 4.** Hypsometry (filled line plots) and ice surface elevation change rates between 2005 and 2008 (crosses), 2008 and 2009 (triangles) at Abraham (ABR), Hidden (HID), and Minaret (MIN) glaciers, and from 2009 to 2011 (circles) at Abraham and Hidden glaciers. Error bars are 1-sigma errors, determined empirically using DEM vertical errors and errors in GPS receiver position. The percentage area surveyed (ratio of the total area of altitude bands containing at least one elevation change measurement to the total glacier area) are given for each time period, following the symbol legend.



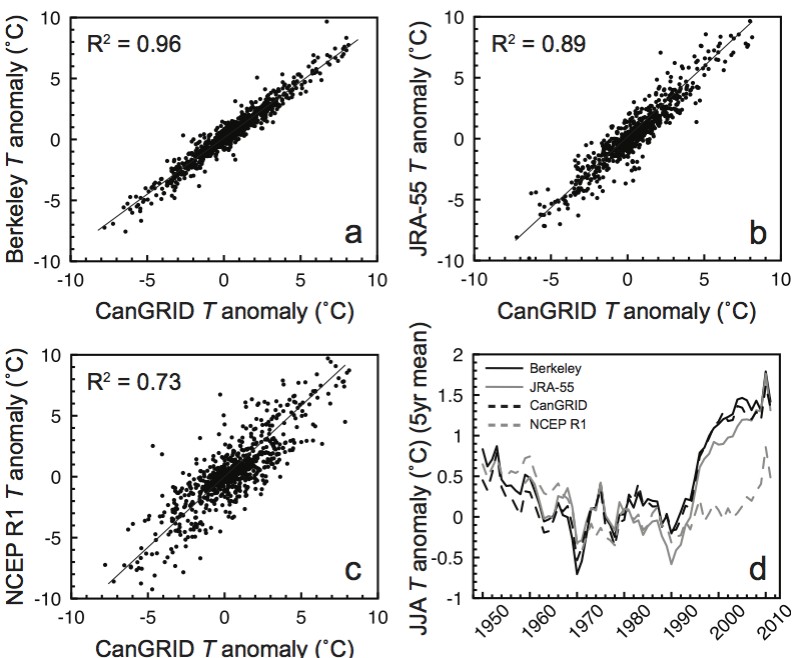

**Figure 5.** CanGRID 2 m air temperature ($T$) anomalies plotted as a function of Berkeley Earth (a), JRA-55 (b), and NCEP R1 (c) anomalies. Solid lines represent linear regression lines. Data were extracted for all available years and averaged across the region 58°N-60°N, 295°E-297°E. Panel d shows smoothed summer (JJA) $T$ anomalies from each dataset, between 1950 and 2011.



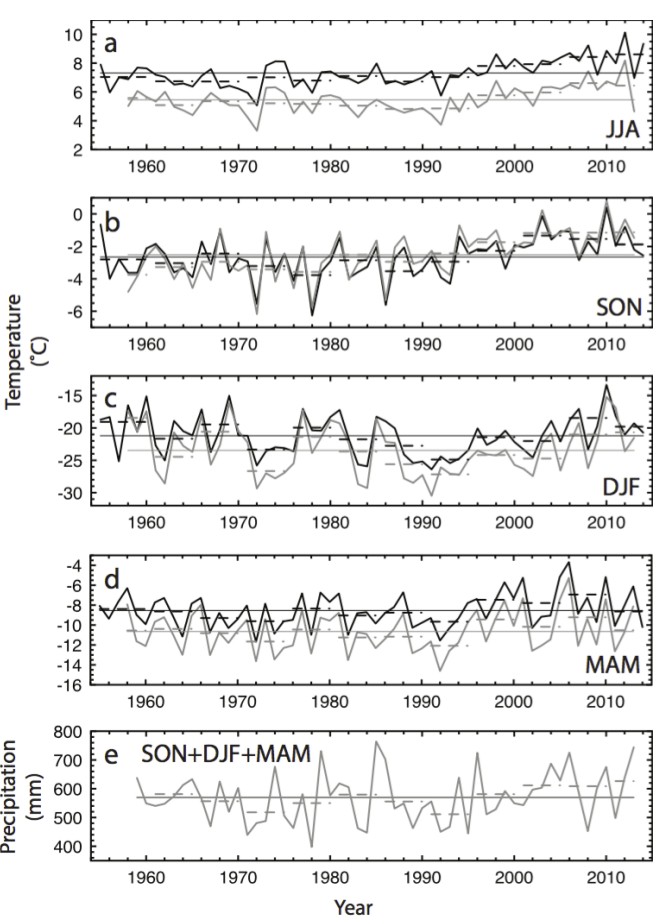

**Figure 6.** Time series of seasonal 2 m air temperatures (a-d), and preceding accumulation season (Sep-May) precipitation totals (e), averaged across the region 58°N-60°N, 295°E-297°E, from Berkeley Earth (black lines) and JRA-55 reanalysis datasets (grey lines). Continuous solid lines are time-series averages and dashed line segments are pentadal means.



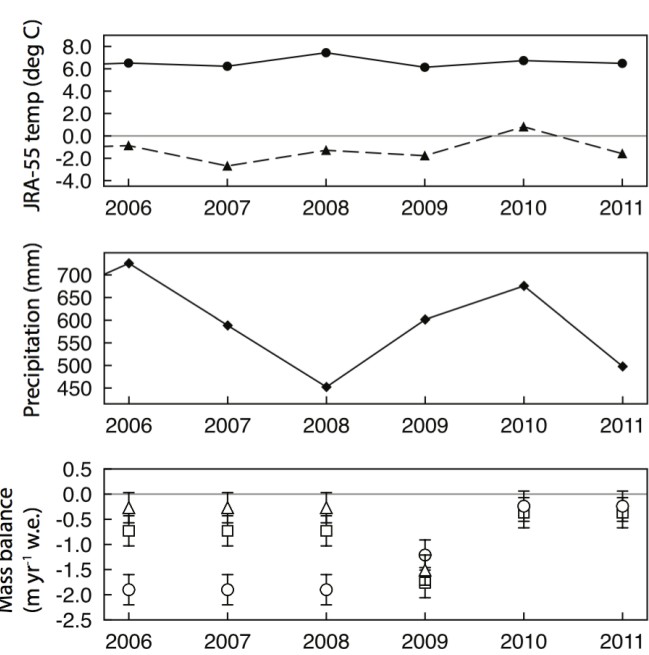

**Figure 7.** Time series of JRA-55 seasonal air temperatures (upper panel: JJA solid line and dots; SON dashed line and triangles), JRA-55 precipitation (middle panel, diamonds), and geodetic mass balances (lower panel) at Abraham (squares), Hidden (circles), and Minaret (triangles) glaciers, between 2005/2006 and 2011. Note that 2005-2008 and 2009-2011 values are annually-averaged, calculated between geodetic measurements at the start and end of both periods.