# Peer review of "Recent changes in area and thickness of Torngat Mountain glaciers (northern Labrador, Canada)"

_The Cryosphere, 2016_

## Referee Comment (RC1) · M. Pelto (Referee) · 7 Oct 2016

Barrand et al (2016) provide a detailed examination of Labrador glacier area, volume and mass balance change. The data set for some aspects is temporally limited. The methods utilized are well explained and effective. Given the lack of temporal mass balance data and finer temporal resolution for area change, the relationships identified to climate are overstated. Most of the changes indicated below are minor. I would encourage use of AAR data when possible. This is an important benchmark study of glacier change in the area, but the climate sensitivity identified is too preliminary to be of significant value.

Specific Comments 1.2: "last" to "only".. Last implies residual from some previous

period.

1.7: List years., could be shorter than as stated.

2.23: icecaps around the GIS and Canadian Arctic Islands. This connection could be better made by citation of Nilsson et al (2015) and Gardner et al (2012).

2.27: Given their small size can the statement of important to Arctic tundra and fjord ecosystem be defended?

2.33: Remove the sentence beginning with… "The majority," This is a distraction that is best left unsaid.

3.68: Reword to make accurate, this is not the first assessment in all the respects noted, since I have previously reviewed two other papers looking at these glaciers in the

4.118: "editing" to "identifying"

5.154: Three different values are listed, which are applied when?

6.194: 2005-2008, 2008-2009 and 2009-2009 should this be changed to 2009-2011?

7.224-230: Reorder the sentences to be chronologic. Begin with 1950 not 2005.

8.242: Move this sentence to make chronologically sequential.

8.255: Worth noting the area change in terms of percent for each of the three glaciers.

8-267: Volume loss appears large compared to resulting area change, deserves a comment.

8.270: Worth noting that Figure 4 illustrates the lack of a sustained accumulation zone for Hidden and Abraham Glacier. Meanwhile Minaret has an accumulation zone, this seems an important distinction.

9.288: This should be put in a Table and largely removed from the text. Figure 7 is not
a substitute for this. Also is there any AAR data that could be added? It is noted that AAR is 0 in some years.

10-313: Sentence needs reworking, hard to discern meeting now.

11.344: how many of the 13 years were above average?

11.345: This indicates PPT trend is inverse to Ba.

11.365: What about Hidden 2005-2008? Examine that one year 2006 and impact of ppt total.

11.373: Which year did the ablation season extend into the Sept.-Nov. period?

11-374: I understand that only preliminary climate-mass balance linkages can be made. However, if no statistical information is provided on these linkages, and this is because the data set is too scant, then I do not think such a linkage can be made.

12.391: Cite Figure 4 in reference to low elevation thinning

12.398: Should this be changed to? These findings suggested that Labrador annual glacier mass balance are controlled by...

Figure 2. Some of the shading issues can be minimized with a bit of photo editing.

Figure 5 and 6 are referenced in the paper after Figure 7

Gardner, A., Moholdt, G., Arendt, A., and Wouters, B.: Accelerated contributions of Canada's Baffin and Bylot Island glaciers to sea level rise over the past half century, The Cryosphere, 6, 1103-1125, doi:10.5194/tc-6-1103-2012, 2012.

Nilsson, J., Sandberg Sørensen, L., Barletta, V. R., and Forsberg, R.: Mass changes in Arctic ice caps and glaciers: implications of regionalizing elevation changes, The Cryosphere, 9, 139-150, doi:10.5194/tc-9-139-2015, 2015.

---

## Referee Comment (RC2) · F. Pálsson (Referee) · 19 Oct 2016

Barrand et al. 2016: In this paper well established methods are used to extract information on glacier mass balance in the Labrador area glaciers, where little has been known to now. The structure of the paper is logical and description of the methods used is clear and easy to read. As often in papers on this topic there could be more rigid justification for error estimates; perhaps using methods similar to i.e. Rolstad 2009 and Magnusson et al. 2016. This would increase significance of the mb results presented, but may be beyond the scope of this work. Given the sparse data available, the attempt to analyse climate sensitivity of the area may be a worthwhile exercise, but is not rigorous enough to add significantly the scientific value of the paper. The mb record however is

an important contribution. (Rolstad, C., Haug, T., and Denby, B.: Spatially integrated geodetic glacier mass balance and its uncertainty based on geostatistical analysis; application to the western Svartisen ice cap, Norway, J. Glaciol., 55, 666-680, 2009. E. Magnússon, J. Muñoz-Cobo Belart, F. Pálsson, H. Ágústsson, and P. Crochet. 2015. Geodetic mass balance record with rigorous uncertainty estimates deduced from aerial photographs and lidar data - Case study from Drangajökull ice cap, NW Iceland.The Cryosphere, 10, 159-177, 2016/www.the-cryosphere.net/10/159/2016/doi:10.5194/tc-10-159-2016)

Below: I have seen M. Pelto comments and agree to all his suggestions and in addition: Line 10. 27% glacier shrinkage: this refers to the area I am sure? not volume?; perhaps adding the word area helps for better clarity Line 12. "negative geodetic mass balance" seems odd when referring to a physical change; I suggest "volume loss"; Lines 13-15. I am not sure if the sparse data allows for such a strong phrasing of the change for control by winter snow variability to control by summer conditions. Consider whether the sentence should be rephrased. Remember that distribution of the winter snow (both from snowfall and redistribution by wind) can play a major role so prevailing wind during winter can be a hidden Joker. 50-53. Slightly negative or slightly positive: I wonder what is the error estimate for the mb for individual glaciers deduced from the in situ survey. Mb spatial variability tends to be high for small valley or cirque glaciers. It would not be surprising that the error is ∼0.5m weq; that is two times the mb values mentioned so maybe close to zero is better. 105. The resultant (resulting?) RMS error of 1950. . . is not a "the" missing before 1950; and the error: is it the difference between the orthoimages and GSPs or.. clarify. 165-179. This is far to detailed, should be boiled down to 2-3 sentences. (All this detail could be included in a supplement document if you find necessary, ). Section 3.1. It would be of great advantage to rewrite this section to chronological order as suggested by M. Pelto. You should also consider changing some of the numbers, taking into account the estimated errors. For example should reduction of 0.66 +-0.41 not be written as 0.7 +-0.4 ?? Do the error estimates allow for stating 0.49% rather than 0.5% or 27% instead of ∼30% and so on. Also consider if

a small volume change (in order of or less than error estimates) and small mb values should be stated as close to zero instead of positive or negative; 3.2 It would help to add a table with all these numbers as suggested by M. Pelto, but a figure similar to the lowest frame of fig 7. or fig 9a in Magnússon et al. (with points w. error bars added for mb of individual years as from the in situ survey) would help the reader to easily grab all this information. This will also help to omit the current reference to fig. 7 prior to that of 5 and 6.
* * *

---

## Author Comment (AC1) · 16 Dec 2016

Barrand et al (2016) provide a detailed examination of Labrador glacier area, volume and mass balance change. The data set for some aspects is temporally limited. The methods utilized are well explained and effective. Given the lack of temporal mass balance data and finer temporal resolution for area change, the relationships identified to climate are overstated. Most of the changes indicated below are minor. I would encourage use of AAR data when possible. This is an important benchmark study of glacier change in the area, but the climate sensitivity identified is too preliminary to be of significant value.

Specific Comments 1.2: "last" to "only".. Last implies residual from some previous period.

*Text amended to read: 'only'.*

1.7: List years., could be shorter than as stated.

*Text amended to read: 'from 1981 to 1984.'*

2.23: icecaps around the GIS and Canadian Arctic Islands. This connection could be better made by citation of Nilsson et al (2015) and Gardner et al (2012).

*Text amended to read: 'and Canadian Arctic archipelago (Nilsson et al., 2015; Gardner et al. 2012)'.*

2.27: Given their small size can the statement of important to Arctic tundra and fjord ecosystem be defended?

*This statement is based on the findings of a comprehensive regional climate change impact assessment report (Brown et al., 2012) cited at the end of the sentence. As currently written, this may give the reader the impression that the 'cultural landscape' comment is referred to by the Brown citation. The text is now amended to place the supporting citation immediately after the statement of importance, to read: 'are an important component of the local Arctic tundra and fjord ecosystem (Brown et al., 2012), and…'*

2.33: Remove the sentence beginning with. . . "The majority," This is a distraction that is best left unsaid.

*Sentence (and corresponding references) is (are) now deleted.*

3.68: Reword to make accurate, this is not the first assessment in all the respects noted, since I have previously reviewed two other papers looking at these glaciers in the

*The reviewer's comment is incomplete but alludes to previous studies investigating these glaciers. As we are unaware of any previous study which has comprehensively examined all Labrador glaciers by means of multiple remote sensing analyses and field-based geodetic measurement, the sentence text is amended to read: 'This study provides the first comprehensive, regional-scale remote-sensing and field-based assessment of the contemporary state of Labrador glaciers…'*

4.118: "editing" to "identifying"

*Text amended to read 'identifying'.*

5.154: Three different values are listed, which are applied when?

*All three sets of scaling parameters are applied to all volume calculations to provide a spread of scaled volumes. The largest difference between calculated volumes is then taken as our measure of uncertainty (following DeBeer & Sharp, 2007; Barrand & Sharp, 2010). The following text is amended to make this clear: 'Total volumes of all glaciers were calculated with the scaling parameters…'.*

6.194: 2005-2008, 2008-2009 and 2009-2009 should this be changed to 2009-2011?

*We thank the reviewer for noticing this typographical error and amend the text to read '2009-2011.'*

7.224-230: Reorder the sentences to be chronologic. Begin with 1950 not 2005.

*We have reordered these sentences chronologically, to read: 'Due to missing photo frames from the 1950s LAB series, only 101 of the 124 glaciers examined in 2005 could be digitized to provide historical glacier areas. In 1950, these 101 glaciers occupied an area of 27.17 km$^2$, shrinking to 19.78 km$^2$ in 2005, then 19.26 km$^2$ in 2008; reductions of 7.39 ± 0.65 km$^2$, or 27% from 1950-2005 (–0.49% yr$^{-1}$), and 0.52 ± 0.33 km$^2$, or 1.37% between 2005 and 2008 (–0.46% yr$^{-1}$). During the three years from 2005 to 2008, the total area of all 124 glaciers in Labrador shrank from 22.46 to 21.80 km$^2$, a reduction of 0.66 ± 0.41 km$^2$, or 3% of the 2005 ice area.'*

8.242: Move this sentence to make chronologically sequential.

*This sentence has now been moved to the end of the paragraph, to make chronologically sequential.*

8.255: Worth noting the area change in terms of percent for each of the three glaciers.

*We have calculated total percentage area changes for each of the Rogerson glaciers, and amend the text to include, as follows: '(-7%)', '(-8%)', '(–12% and –11%, respectively)', '(–14% and –13%, respectively)', '(-21%)', '(-23%)'.*

8-267: Volume loss appears large compared to resulting area change, deserves a comment.

*We include the following amended text: 'These very large thinning rates (compared to resultant area changes' decreased further up-glacier…'. The following sentences, comparing thinning rates between shaded / non-shaded, greater / lesser debris-covered ice provide some possible explanations for these different thinning rates between glaciers.*

8.270: Worth noting that Figure 4 illustrates the lack of a sustained accumulation zone for Hidden and Abraham Glacier. Meanwhile Minaret has an accumulation zone, this seems an important distinction.

*We amend the text to read: '…Hidden Glacier, and no sustained accumulation zone. In contrast, all of the ice at Minaret Glacier exists at…'. This point is also discussed later in the manuscript, at the paragraph beginning at line 283.*

9.288: This should be put in a Table and largely removed from the text. Figure 7 is not a substitute for this. Also is there any AAR data that could be added? It is noted that AAR is 0 in some years.

*It is the opinion of the authors that the small amount of yearly mass balance data are appropriately presented in text form. A similar comment from the scientific editor led to these data also being presented in Figure 7 in a pre-discussion paper revision. The addition of such a small table of these data, alongside the figure, would not add much to the paper. The figure serves a dual purpose, illustrating the short mass balance time series, and providing a qualitative link to prevailing climate variables during mass balance years since 2006. Unfortunately, accumulation area ratio data are not available from our geodetic measurements. The AAR = 0 point is discussed further down: '…between 2005 and 2008, and 2008 and 2009 coincided with with nonexistent or very small accumulation areas at these glaciers.'*

10-313: Sentence needs reworking, hard to discern meeting now.

*We thank the reviewer for bringing this text processing error to our attention. The text was included by mistake during pre-discussion paper revision. The text has now been removed.*

11.344: how many of the 13 years were above average?

*Text amended to read: '…shows 8 years of above average precipitation between 1974 and 1986…'.*

11.345: This indicates PPT trend is inverse to Ba.

*There is a small increasing trend in precipitation post 1990 which occurs together with strong atmospheric warming (Figure 6). However, it is difficult to draw any strong conclusions on precipitation – mass balance links as there are several low precipitation total years, and the 1990-2015 climate data overlap with mass balance observations from 2006-2011 only (a period which includes several very low precipitation yearly totals in 2008 and 2011 (Figure 6e).*

11.365: What about Hidden 2005-2008? Examine that one year 2006 and impact of ppt total.

*We amend the text to include: 'Larger negative balances at Hidden glacier during this time period may suggest greater insensitivity to precipitation.'.*

11.373: Which year did the ablation season extend into the Sept.-Nov. period?

*The text is amended to include: '…with melting conditions extending into the September-November period during 2010 (Figure 6b)'.*

11-374: I understand that only preliminary climate-mass balance linkages can be made. However, if no statistical information is provided on these linkages, and this is because the data set is too scant, then I do not think such a linkage can be made.

*We agree that a formal attribution of climate changes on Labrador glacier mass balance probably cannot be made with such sparse data. Therefore, we include the qualifier 'may' into this sentence, to read: '…inter-annual variability in winter precipitation and post-1995 climate warming may play important roles in contemporary Labrador glacier mass changes'.*

12.391: Cite Figure 4 in reference to low elevation thinning

*Text amended to read: '…between 2005 and 2011 (Figure 4), mostly..'.*

12.398: Should this be changed to? These findings suggested that Labrador annual glacier mass balance are controlled by. . .

*As above, this sentence text is amended as follows: '…glacier mass trends may be controlled by variability in winter precipitation, and increasingly by…'.*

Figure 2. Some of the shading issues can be minimized with a bit of photo editing.

*We are unsure as to the rendering of Figure 2 in the reviewer's manuscript copy, but have previously adjusted the image contrast and opacity settings to adequately view shaded areas. We suggest (and are happy to enter into) a dialogue with journal copy editors to produce the most viewer-friendly contrast settings for Figure 2.*

Figure 5 and 6 are referenced in the paper after Figure 7

*The first two references to Figure 7 have now been removed so that Figures 5, 6 and 7 now occur in chronological order.*

Gardner, A., Moholdt, G., Arendt, A., and Wouters, B.: Accelerated contributions of Canada's Baffin and Bylot Island glaciers to sea level rise over the past half century, The Cryosphere, 6, 1103-1125, doi:10.5194/tc-6-1103-2012, 2012.

Nilsson, J., Sandberg Sørensen, L., Barletta, V. R., and Forsberg, R.: Mass changes in Arctic ice caps and glaciers: implications of regionalizing elevation changes, The Cryosphere, 9, 139-150, doi:10.5194/tc-9-139-2015, 2015.

*Both of these references are now included in the reference list.*
Barrand et al. 2016: In this paper well established methods are used to extract information on glacier mass balance in the Labrador area glaciers, where little has been known to now. The structure of the paper is logical and description of the methods used is clear and easy to read. As often in papers on this topic there could be more rigid justification for error estimates; perhaps using methods similar to i.e. Rolstad 2009 and Magnusson et al. 2016. This would increase significance of the mb results presented, but may be beyond the scope of this work.

*This is an interesting suggestion, but one that we agree is beyond the scope of this work. The Rolstad semivariogram approach relies on good quality elevation differences from surrounding non-ice terrain, of which we have from only one of our geodetic datasets (2005 Parks Canada DEM). We should mention here, however, that following the scientific editor's pre-discussion paper review comments, we have taken account of weakly positive spatial autocorrelation in our geodetic measurements, adjusting the square root divisor term in the error (quadrature) equation by a smaller number of independent measurements (one per 10 m elevation bin, see Section 2.4).*

Given the sparse data available, the attempt to analyse climate sensitivity of the area may be a worthwhile exercise, but is not rigorous enough to add significantly the scientific value of the paper.

*We agree, and on the basis of similar review comments from referee 1, have amended the text to include the important 'may' qualifier (see details above).*

The mb record however is an important contribution. (Rolstad, C., Haug, T., and Denby, B.: Spatially integrated geodetic glacier mass balance and its uncertainty based on geostatistical analysis; ap- plication to the western Svartisen ice cap, Norway, J. Glaciol., 55, 666-680, 2009. E. Magnusson, J. Munõz-Cobo Belart, F. Palsson, H. Agustson, and P. Crochet. 2015. Geodetic mass balance record with rigorous uncertainty estimates deduced from aerial photographs and lidar data - Case study from Drangajokull ice cap, NW Iceland.The Cryosphere, 10, 159-177, 2016/www.the-cryosphere.net/10/159/2016/doi:10.5194/tc- 10-159-2016)

Below: I have seen M. Pelto comments and agree to all his suggestions and in addition: Line 10. 27% glacier shrinkage: this refers to the area I am sure? not volume?; perhaps adding the word area helps for better clarity

*We would argue that the phrase 'regional glacier shrinkage' should refer to glacier area rather than volume changes. However, to avoid any further confusion for readers, we amend this text to read: '…regional glacier area loss of 27%...'.*

Line 12. 'negative geodetic mass balance' seems odd when referring to a physical change; I suggest 'volume loss';

*We amend the text to read: '…and volume losses at Abraham, Hidden and…'.*

Lines 13-15. I am not sure if the sparse data allows for such a strong phrasing of the change for control by winter snow variability to control by summer conditions. Consider whether the sentence should be rephrased. Remember that distribution of the winter snow (both from snowfall and redistribution by wind) can play a major role so prevailing wind during winter can be a hidden Joker.

*We have amended this sentence text based on similar review comments from referee 1.*

50-53. Slightly negative or slightly positive: I wonder what is the error estimate for the mb for individual glaciers deduced from the in situ survey. Mb spatial variability tends to be high for small valley or cirque glaciers. It would not be surprizing that the error is ∼0.5m weq; that is two times the mb values mentioned so maybe close to zero is better.

*We would very much have liked to have known the error estimates of mass balance observations from the work of Rogerson et al. (1986) - unfortunately this information was unavailable. We agree that there is a possibility that uncertainties may be large (masking even the sign of mass balance for these years) and adjust the manuscript text here to read: '…in 1983 (uncertainties are unreported)'.*

105. The resultant (resulting?) RMS error of 1950. . . is not a 'the' missing before 1950; and the error: is it the difference between the orthoimages and GSPs or.. clarify.

*We have amended this sentence to read: 'The resulting root mean square (RMS) error of the 1950 orthophotos…'. The error describes a goodness of fit between orthoimages and ground control points per image frame. The text is amended to read: 'The resulting root mean square (RMS) error between the 1950 orthoimages and GCPs was <5 m across all images.'*

165-179. This is far to detailed, should be boiled down to 2-3 sentences. (All this detail could be included in a supplement document if you find necessary, ).

*We have removed two sentences from this section to provide greater brevity. The sentences removed referred to base-station precise point position processing and correction of survey positions. These edits reduce this paragraph section from 7 sentences to 5, albeit keeping important details about processing steps, positional data selection and measurement uncertainty.*

Section 3.1. It would be of great advantage to rewrite this section to chronological order as suggested by M. Pelto.

*This has now been done following the comments of referree 1 (see above).*

You should also consider changing some of the numbers, taking into account the estimated errors. For example should reduction of 0.66 +-0.41 not be written as 0.7 +-0.4 ?? Do the error estimates allow for stating 0.49% rather than 0.5% or 27% instead of ∼30% and so on.

*We have edited this text to reduce the number of significant figures of area change values from 2 to 1, following the reviewer's suggestion. This is now consistent with the reporting of uncertainties.*

Also consider if a small volume change (in order of or less than error estimates) and small mb values should be stated as close to zero instead of positive or negative;

*We agree and have added the following sentence to Section 3.2: 'Considering uncertainties, small mass balance values from 2010-2011 may be considered as close to zero rather than negative.'.*

3.2 It would help to add a table with all these numbers as suggested by M. Pelto, but a figure similar to the lowest frame of fig 7. or fig 9a in Magnússon et al. (with points w. error bars added for mb of individual years as from the in situ survey) would help the reader to easily grab all this information. This will also help to omit the current reference to fig. 7 prior to that of 5 and 6.

*Following our response to referee 1, we feel that a Table would be a poor use of journal space (given the small number of mass balance values), and that this information is adequately communicated in this short text and with later reference to Figure 7 (see response to comments above).*

*Thanks again, Etienne, Mauri and Finnur, for your substantive and helpful comments. We amend the following sentence to the manuscript acknowledgements, to read: 'We thank the editor Etienne Berthier and referees Mauri Pelto and Finnur Pálsson for comments which improved the manuscript.'.*

*Nick*

---

## Author Response (AR2)

**tc-2016-171 Response to final editor reviews (12-12-2016)**

*We thank the scientific editor for his thoughtful and thorough final review of our paper. We have revised the manuscript to address your review comments (see below). Throughout this response to review document your (editor review) comments are provided in regular, non-italic font text, our response comments are provided in italic text (as here), and any changes to the revised manuscript text are provided in quotation marks and italic, blue font text. Along with this response to referee review document, we will upload to the The Cryosphere website a revised manuscript file with marked-up changes (text edits provided in italic, blue font text, as in this document), and a non-marked-up, final revised manuscript file (all text in black non-italic font).*

*We hope these final revisions will be sufficient to warrant acceptance for publication in The Cryosphere.*

*Thanks again, and on behalf of the authors,*
*Nick Barrand*

**Editor Decision: Publish subject to minor revisions (Editor review)** (21 Dec 2016) by Dr. Etienne Berthier. Comments to the Author:

L13-14. Both reviewers and I agreed that the link between climate fluctuations and mass balance is currently not really demonstrated. "appear" suggest this in the abstract but I also ask you to add a statement clarifying that this link is based on very thin data and need further work: "However, further measurements and analysis are required to fully understand the mass balance sensitivity of Labrador glaciers". (feel free to use a different wording but this is the idea)

*The manuscript text in the abstract is amended to read: 'Glacier mass balances appear to be controlled by variations in winter precipitation and, increasingly, by strong summer and autumn atmospheric warming since the early-1990s, though further observations are required to fully understand mass balance sensitivities.'.*

*The manuscript conclusion is also now amended to read: 'These findings suggested that Labrador glacier mass trends may be controlled by variability in winter precipitation, and increasingly by strong summer and autumn atmospheric warming since the early-1990s, though further observations are required to confirm these linkages.'.*

L44. "w.e." not defined

*Text amended to read: '…water equivalent (w.e.).'*

L45. Glacier missing here I think after "Superguksoak"

*Text amended to read: 'Superguksoak Glacier…'.*

L48. 1986a? Check the correct call for the Rogerson-1986 references.

*Text amended to read: '(Rogerson, 1986b)'.*

L200. Currently authors used a constant density of 918 kg/m3 to convert volume to mass. No error is associated to this choice and the use of this value has not been justified. Such a pure ice density (I though pure ice density was 917 kg/m3 but maybe I am wrong here) would be only justified if NO firn was present over the glaciers during ALL surveys. Is it the case? Otherwise a lower density should be applied as justified by Huss, M.: Density assumptions for converting geodetic glacier volume change to mass change, The Cryosphere, 7(3), 877–887, doi:10.5194/tc-7-877-2013, 2013. In all cases, an uncertainty should be added.

*Field geodetic surveys in 2008 and 2009 took place over mostly bare glacier ice with very little or no snow or firn present (AAR = 0, see earlier response to review comments). However, we cannot be completely certain of no change in the firn density profile (Sorge's Law). Also, we cannot be certain that the 2005 DEM and 2011 geodetic surveys were collected from snow and firn-free ice conditions (or that the density-depth profile was time invariant between these surveys). For these reasons, and based on the editor's recommendation, we have recalculated 2005-2008, 2008-2009 and 2009-2011 geodetic mass balances based on the lower 'wide range of conditions' density conversion of Huss (2013) (850 +/- 60 kg m³). This approach also has the advantage of introducing an error term to the volume-mass conversion, which we now incorporate into our total error budget. The manuscript text of this section is now amended to read: ‘Volume changes were converted to geodetic mass balances in water equivalent units (m yr⁻¹ w.e.) by dividing by the glacier area, assuming an ice density of 850 ± 60 kg m⁻³ (Huss, 2013), and in the case of 2005-2008 and 2009-2011 measurements, dividing by the time between epochs.’.*

*These recalculations have the result of producing slightly less negative mass balances, with slightly larger uncertainties. Each of these amended values are now included in the manuscript text (in Sections 3.2 onwards) and in a replotted Figure 7 - adjustments that do not alter the conclusions of the manuscript.*

L271. "with the largest changes": authors should write "thinning" here. Otherwise the positive values in the parenthesis that follow could be misinterpreted as a thickening...

*To differentiate between simply ‘thinning’ and ‘the largest changes’ (i.e. the greatest thinning), the text is amended here to read: ‘…with substantial thinning (\sim 4 m yr⁻¹)…’.*

L290. authors can skip geodetic but not "mass" here.

*The word ‘geodetic’ has now been deleted.*

L358. The use of "preceding" here and also earlier (L328) ambiguous. A mass balance year is made of an accumulation season followed by an ablation season. Thus, the "preceding" accumulation season could be mis-interpreted as the one from the previous mass balance year. Avoid such ambiguity.

*At line 328 (former), the words ‘the preceding years’ are deleted.*
*At line 359 (formed), the word ‘preceding’ is now deleted.*

L388. Delete (Figure 4). No need to call a figure in the conclusion

*Reference to Figure is now deleted from conclusions section ‘(Figure 4)’.*

L391. Your acceleration in the thinning rate and the very negative mass balances after 2005 are in line with the findings of Papasodoro et al. for the nearby (300 km further north) Terra

Nivae ice cap (http://www.the-cryosphere.net/9/1535/2015/). You may thus compare yours and their value. It would give a more regional scope to your conclusions.

*We thank for the editor for bringing this study to our attention. We include the following text at lines 303-305 (in Section 3.2): 'This strongly negative post-2005 mass balance is similar in magnitude to that measured at nearby Terra Nivea ice cap on southern Baffin Island (Papasodoro et al., 2015).'.*

*We also include the following text in the conclusions section: '…being the most negative mass balance yet recorded at this or any other glacier in Labrador, and similar in magnitude to that measured at neaby Terra Nivea ice cap.'.*

Legend of Figure 2. "base stations" (space missing I think)

*Text amended to read: '…base stations…'.*

Non-public comments to the Author:
Nick and co-authors,
No Christmas gift sorry. No yet. Just a small effort and this should work.
Etienne

*Thanks again for your time and efforts reviewing this manuscript, Etienne.*
*Nick*